Extending the fossil record of late Oligocene non-biting midges (Chironomidae, Diptera) of New Zealand

http://orcid.org/0000-0003-1893-3215 Baranov Viktor O. 1 viktor.baranov@ebd.csic.es
http://orcid.org/0000-0002-6744-6811 Hammel Jörg U. 2
Lee Daphne E. 3
Schmidt Alexander R. 4
http://orcid.org/0009-0007-6858-8612 Kaulfuss Uwe 5
1 Estación Biológica de Doñana, Consejo Superior de Investigaciones Científicas , Sevilla , Spain
2 Institute of Materials Physics, Helmholtz-Zentrum Hereon , Geesthacht , Germany
3 Department of Geology, University of Otago , Dunedin , New Zealand
4 Department of Geobiology, University of Göttingen , Göttingen , Germany
5 Department of Animal Evolution & Biodiversity, University of Göttingen , Göttingen , Germany
Żyła Dagmara
Electronic publication date: 2025 Feb 21
Publication date: 2025
Volume: 13
Electronic Location ID: e18893
Received 2024 Sep 6; Accepted 2025 Jan 2
Copyright: © 2025 Baranov et al.
Copyright year: 2025
Copyright holder: Baranov et al.
License: This is an open access article distributed under the terms of the Creative Commons Attribution License, which permits unrestricted use, distribution, reproduction and adaptation in any medium and for any purpose provided that it is properly attributed. For attribution, the original author(s), title, publication source (PeerJ) and either DOI or URL of the article must be cited.
License URL: https://creativecommons.org/licenses/by/4.0/

Keywords: Chironomidae, Fossil insects, Palaeoecology, Pomahaka Formation, Zealandia

Funding: Marsden Grant UOO1416 Royal Society of New Zealand Spanish State Agency for Innovation’s Ramon y Cajal fellowship RyC2021-032144-I German Research Foundation 429296833 Open Access Publication Support Initiative from the CSIC Unit of Information Resources for Research (URICI) DESY Block Allocation Group project “Scanning the past—Reconstructing the diversity in million years old fossil amber specimens using SRµCT” at PETRA III Our results build on the earlier project “Captured in amber: ecological complexity in New Zealand’s ancient araucarian forests”, funded by a Marsden Grant (UOO1416) by the Royal Society of New Zealand. This study was funded by the Spanish State Agency for Innovation’s Ramon y Cajal fellowship (RyC2021-032144-I), project title “Climate change in the past and present & Insect decline” (to Viktor Baranov) and by the German Research Foundation (DFG project 429296833 to Uwe Kaulfuss), project title “Palaeontology, biogeography and evolution of New Zealand insects”. Open Access Publication Support Initiative from the CSIC Unit of Information Resources for Research (URICI). Scanning of the specimens was supported by the DESY Block Allocation Group project “Scanning the past-Reconstructing the diversity in million years old fossil amber specimens using SRµCT” at PETRA III. The funders had no role in study design, data collection and analysis, decision to publish, or preparation of the manuscript.

==============================
Background

The modern chironomid fauna of New Zealand is diverse, highly endemic and reflects a complex biogeographical history. This fauna has been important for developing phylogenetic and biogeographic concepts including Brundin’s writings on transantarctic relationships but until now the fossil record to support these reconstructions has been very limited. Here we describe the first fossil species of Chironomidae, subfamily Orthocladiinae, from New Zealand, based on inclusions in amber from the late Oligocene Pomahaka Formation of the South Island.

Methods

We examined newly excavated fossil tree resin (amber) from the late Oligocene Pomahaka Formation in southern New Zealand for inclusions. Amber pieces containing chironomids were prepared and morphologically investigated using light-microscopy and µCT-scanning. Specimens were taxonomically evaluated using identification keys for modern adult chironomid midges. Habitus and key morphological features of each specimen were documented photographically and/or by line drawings.

Results

Thirteen Chironomidae specimens from Pomahaka amber were identified as members of the subfamily Orthocladiinae Kieffer. Bryophaenocladius zealandiae sp. nov. Baranov is the first Southern Hemisphere fossil of the genus. Bryophaenocladius Thienemann, 1934 is absent from the extant fauna of the main islands of New Zealand; however, it may be present on the subantarctic Auckland Islands. Two incompletely preserved specimens are described as Morphotype 1 cf. Bryophaenocladius zealandiae. Based on a male adult, Pterosis extinctus sp. nov. Baranov is described as the first fossil record of the extant genus Pterosis Sublette and Wirth, today represented by a single endemic species on the New Zealand subantarctic Auckland Islands and Campbell Island. Two female adult specimens are described as Morphotype 2 cf. Metriocnemini. The new fossils of the genera Bryophaenocladius and Pterosis belong to chironomid taxa requiring terrestrial or semi-aquatic habitats for larval development, supporting the notion of a humid forest swamp paleoenvironment for the Pomahaka amber source forest.

Introduction

Non-biting midges (Chironomidae) have historically served as a model group for the development of both modern phylogenetic analysis and historical biogeography (Hennig, 1960; Brundin, 1966). Studies of the extant Chironomidae fauna of New Zealand have played a major role in understanding transantarctic vicariance patterns (Brundin, 1966; Krosch & Cranston, 2013). In particular, phylogenetic studies of the Podonominae, southern Diamesinae and austral Orthocladiinae were seminal for understanding vicariance patterns caused by the break-up of Gondwana (Brundin, 1966; Krosch et al., 2011; Krosch & Cranston, 2013). The fossil record of New Zealand’s Chironomidae fauna is therefore very important for understanding biogeographic patterns in the Southern Hemisphere (Schmidt et al., 2018; Baranov, Haug & Kaulfuss, 2024).

Our knowledge of the fossil history of Chironomidae in New Zealand has been very limited, so far, despite significant studies mentioned above. Schmidt et al. (2018) reported four specimens of Bryophaenocladius Thienemann (Orthocladiinae) from Oligocene amber from the South Island, which are included in our descriptions herein. Baranov, Haug & Kaulfuss (2024) described three morphotypes of immature Chironomidae from Early Miocene lake sediments at Foulden Maar on the South Island. Subfossil records of Chironomidae include the larvae of Corynocera duffi Deevey, 1955 from Holocene swamp deposits in Canterbury, South Island (Deevey, 1955) and numerous other chironomid taxa identified from various Holocene sites on South Island (Schakau, 1991; Woodward & Shulmeister, 2007; Dieffenbacher-Krall et al., 2008).

Considering this limited fossil record, it is difficult to improve our understanding of the evolutionary history of Chironomidae in New Zealand, particularly for the subfamily Orthocladiinae. Thus, any additional deep time records of this subfamily from New Zealand are of great value. In this study, we describe two new species of Orthocladiinae from Oligocene amber from the South Island of New Zealand. These new discoveries add valuable knowledge to our understanding of the past diversity and historical biogeography of Orthocladiinae in New Zealand.

Geological setting

The Chironomidae specimens studied here are inclusions in amber from the estuarine late Oligocene Pomahaka Formation in southern New Zealand. Fossiliferous amber was collected from a lignite bed and associated carbonaceous mudstone in a temporary excavation pit on private farmland near Pomahaka River approx. 12 km south of Tapanui (46.04450°S, 169.22292°E) (Fig. 1). The locality is registered as G45/f0107 in the New Zealand Fossil Record File (GNS Science & Geological Society of New Zealand, 2024). Fourier-transform infrared spectroscopy analysis of amber from the site indicates an araucarian, Agathis-like parent plant, which is supported by finds of araucarian wood and abundant pollen of Araucariacites australis Cookson, 1947 in Pomahaka Formation sediments (Pocknall, 1982; Lee et al., 2009; Kaulfuss et al., 2024). Within the lignite and underlying carbonaceous mudstones, amber is very common and occurs randomly distributed as mm-sized droplets to dm-sized lumps and blocks, showing no signs of sorting and abrasion by reworking and transport. Combined with reconstructions from sedimentological and palynological data (Pocknall, 1982; Lindqvist, Gard & Lee, 2016), this indicates in situ resin deposition and amber formation in a domed forest swamp adjacent to a brackish mire or saltmarsh within an estuarine paleoenvironment. A comprehensive facies analysis of Pomahaka Formation was published by Lindqvist, Gard & Lee (2016). The late Oligocene age (Chattian, New Zealand stage Duntroonian, 27.3–25.3 Ma) for the Pomahaka Formation has been established on palynomorph and molluscan biostratigraphy (Wood, 1956; Pocknall, 1982; Beu & Maxwell, 1990).

Figure 1 Late Oligocene Pomahaka amber.

(A) Map of amber locality G45/f0107 near Tapanui, southern New Zealand. (B) Typical appearance of layered, fossiliferous Pomahaka amber. (C) Epoxy-embedded pieces of Pomahaka amber with newly discovered Chironomidae inclusions.

Materials and Methods

Material

Thirteen Chironomidae inclusions from Pomahaka amber were studied herein, including four specimens reported as Bryophaenocladius Thienemann, 1934 or closely related to it by Schmidt et al. (2018) and nine newly discovered specimens. Three of the Bryophaenocladius specimens reported by Schmidt et al. (2018) are fossilized in a single piece of amber and a further specimen in a separate piece. The collection number No. OU35028.2 collectively assigned to all four specimens in Schmidt et al. (2018) is here replaced by individual numbers for each specimen (Nos. OU47579–OU47582). The nine new specimens originate from a single amber piece made up of multiple thin layers formed by successive resin flows but were separated and prepared as individual pieces (Figs. 1B, 1C). The type material and associated specimens are deposited in the Geology Museum of the Geology Department, University of Otago (OU); collection numbers are provided below in the Systematic Paleontology section.

Preparation and imaging

Layered pieces of amber were microscopically examined for biological inclusions and subsequently separated along surfaces of individual resin flows. In instances where this resulted in the exposure of wings at the surface of the amber piece, wings were photographed with a binocular stereomicroscope (Carl Zeiss Stemi 508 with a Canon EOS 70D digital camera) prior to further preparation. Where possible, the thin and brittle, inclusion-bearing amber shards were ground and polished to obtain dorsal, ventral and/or lateral views of inclusions. Polished amber shards, and those too small and fragile for polishing, were embedded in epoxy resin to stabilise specimens, applying the protocol provided by Sadowski et al. (2021). Epoxy-embedded amber pieces were ground and polished using a grinder/polisher machine (Buehler Eco-Met 250) and CarbiMet silicon carbide abrasive papers (CarbiMet) and/or manually using a set of wet silicon carbide abrasive papers (FEPA P #220–4000).

Specimens were studied with a Carl Zeiss AxioScope A1 compound microscope and photographed with a Canon 5D digital camera. Figures were generated with Helicon Focus (8.2.0) software and enhanced using Adobe® Photoshop CC. Line drawings were prepared with Inkscape 1.1 software.

µCT-scanning

Two Chironomid specimens preserved in close proximity in nearly opaque amber could not be separated and studied by light microscopy. These specimens (No. OU47580, No. OU47581) were scanned on the Imaging Beamline P05 (Lytaev et al., 2014) operated by the Helmholtz-Zentrum Hereon at the PETRA III storage ring (Deutsches Elektronen Synchrotron—DESY, Hamburg, Germany), using a photon energy of 18 keV and a sample-to-detector distance of 100 mm. Projections were recorded with a custom 20 MP CMOS imaging system with an effective pixel size of 1.28 µm (Lytaev et al., 2014). For each tomographic scan, 3,601 projections were recorded at equal intervals between 0 and π. Reconstruction was carried out by applying a transport of intensity phase retrieval approach and using the filtered back projection algorithm (FBP). This workflow was carried out in a custom reconstruction pipeline using MATLAB (Math-Works) and the Astra Toolbox (Moosmann et al., 2014; Van Aarle et al., 2015, 2016). Raw projections were binned twice for further processing, resulting in an effective pixel size of the reconstructed volume (voxel) of 2.56 µm. Scanned volumes were reconstructed using Drishti ver. 2.6.6 (Limaye, 2012). To decrease the demands for computer memory, we converted all stacks into eight-bit tiffs, downscaled all tiffs by 50% and subsequently cropped the empty space around the amber piece using Fiji ‘scale’ and ‘crop’ functions (Schindelin et al., 2012). Volumes were rendered in Drishti ver. 2.6.6 (Limaye, 2012).

Terminology and taxonomy

Our morphological terminology is based on Sæther (1980) and Marshall et al. (2017). Specimens were evaluated using the keys provided by Freeman (1959), Sæther (1973, 1977, 1983), Albu (1974), Sublette & Wirth (1980), Pinder & Armitage (1986), Armitage (1987), Cranston, Oliver & Sæther (1989), Willassen (1996), Andersen & Schnell (2000), Wang, Sæther & Andersen (2001), Kaczorowska & Giłka (2002), Wang, Liu & Epler (2004), Makarchenko & Makarchenko (2006), Wang, Andersen & Sæther (2006), Langton & Pinder (2007), Du, Wang & Sæther (2011), Hazra & Das (2011), Epler (2012), Lin, Qi & Wang (2012), Moubayed & Lods-Crozet (2022), and Moubayed & Langton (2023).

Leg measurements of specimens are mainly approximated values only, due to the difficulty of measuring the variously oriented legs in the amber.

The electronic version of this article in Portable Document Format (PDF) will represent a published work according to the International Commission on Zoological Nomenclature (ICZN), and hence the new names contained in the electronic version are effectively published under that Code from the electronic edition alone. This published work and the nomenclatural acts it contains have been registered in ZooBank, the online registration system for the ICZN. The ZooBank LSIDs (Life Science Identifiers) can be resolved and the associated information viewed through any standard web browser by appending the LSID to the prefix http://zoobank.org/. The LSID for this publication is: urn:lsid:zoobank.org:pub:1B39CC4B-AA24-4D9F-819B-F1834E615C5C. The online version of this work is archived and available from the following digital repositories: PeerJ, PubMed Central SCIE and CLOCKSS.

Results

Systematic Paleontology

Order Diptera Linnaeus, 1758

Family Chironomidae Newman, 1834

Subfamily Orthocladiinae Kieffer, 1911

Genus Bryophaenocladius Thienemann, 1934

Bryophaenocladius zealandiae sp. nov. Baranov

(Figs. 1C, 2–7; Table 1)

Figure 2 Bryophaenocladius zealandiae sp. nov. Baranov, holotype OU47576.

(A) Habitus, dorsal view. (B) Habitus, ventral view. (C) Antenna, ventral view. (D) Tibial spurs. Abbreviations: Ac, Acrostichal setae; Dc, dorsocentral setae; Ti1, foreleg tibia; Ti2, midleg tibia; Ti3, hindleg tibia.

Figure 3 Bryophaenocladius zealandiae sp. nov. Baranov, wing of holotype OU47576.

(A) Photomicrograph. (B) Line drawing. Abbreviations: An, anal vein; B, brachiolum; C, costal vein; ce, costal extension; Cu1, cubital vein 1; Cu2, cubital vein 2; M1+2, medial vein 1+2; M3+4, medial vein 3+4; R1, radial vein 1; R2+3, radial vein 2+3; R4+5, radial vein 4+5; RM, radial medial crossvein; Sc, subcostal vein; Sq, squama.

Figure 4 Bryophaenocladius zealandiae sp. nov. Baranov, hypopigium of holotype OU47576.

(A) Photomicrograph, dorsal. (B) Line drawing, dorsal. (C) Photomicrograph, ventral. (D) Line drawing, ventral. Abbreviations: AP, anal point; Gc, gonocoxite; Gs, gonostylus; IVo, inferior volsella; TVIII, abdominal tergite 8; TIX abdominal tergite 9.

Figure 5 Bryophaenocladius zealandiae sp. nov. Baranov, paratypes.

(A, B) Habitus and hypopygium of paratype OU47540. (A1) Hindtibia spurs. (C, D) Hypopygium and habitus of paratype OU47575. (E) Habitus of paratype OU47572.

Figure 6 Bryophaenocladius zealandiae sp. nov. Baranov, associated specimen OU47579.

(A) Habitus. (A1) Overview of the amber piece containing the specimen. (B) Habitus, opposite side of the body.

Figure 7 Bryophaenocladius zealandiae sp. nov. Baranov, CT scans of associated specimens.

(A) Habitus of specimens OU47580 and OU47581 in the same amber piece. (B) Habitus of specimen OU47580, dorso-lateral view. (C) Hypopygium of specimen OU47581. (D) Hypopygium (OU47581), marked. Abbreviations: AP, anal point; Gc, gonocoxite; Gs, gonostylus; TIX, abdominal tergite 9. (E) Overview of the amber piece containing specimens OU47580, OU47581 and OU47582.

Table 1 Length (in μm) of leg segments of Bryophaenocladius zealandiae sp. nov. Baranov, males (measured on different numbers of specimens, depending on the preservation of the leg segments of the fossil).

Leg	Femora	Tibia	Ta1	Ta2	Ta3	Ta4	Ta5	
Foreleg	255–547	255–570	136–260	58–100	50–84	36–45	50–65	
350 (n = 5)	370 (n = 5)	206 (n = 3)	78 (n = 3)	67 (n = 3)	41 (n = 3)	55 (n = 3)	
Midleg	255–543	248–656	136–287	66–145	67–82	57–88	44–84	
400 (n = 6)	374 (n = 4)	202 (n = 3)	96 (n = 3)	76 (n = 3)	70 (n = 3)	63 (n = 3)	
Hindleg	305–541	290–607	202–459	81–177	85–111	45–73	43–60	
419 (n = 6)	454 (n = 6)	286 (n = 6)	131 (n = 5)	100 (n = 5)	57 (n = 5)	52 (n = 4)	
Note:

Values are given as min–max range and mean.

Zoobank LSID: urn:lsid:zoobank.org:act:2FE3D4EA-5F0B-4BB4-A2CC-E88557CE825

Holotype. No. OU47576, adult male, complete specimen in a piece of translucent, yellowish-orange amber with dimensions of 8 × 4 × 0.5 mm. Head and thorax covered by cloudy coating ventrally and dorsally, and parts of the thorax and abdomen obscured by numerous bubbles (Figs. 1C, 2–4).

Paratypes. No. OU47540, No. OU47572 and No. OU47575, adult males, generally well preserved but some morphology obscured by air bubbles (Fig. 5)

Associated specimens. No. OU47579 in a semi-translucent piece, No. OU47580, No. OU47581 and No. OU47582 together in one nearly opaque piece, mostly obscured by detritus and air bubbles; all adult males (Figs. 6, 7).

Derivation of name. The specific epithet refers to the largely submerged continent Zealandia.

Type locality and horizon. Temporary lignite pit, site G45/f0107, near Tapanui, southern New Zealand; Pomahaka Formation, late Oligocene (Chattian, New Zealand stage Duntroonian).

Diagnosis. The new species can be easily distinguished from any living and fossil Bryophaenocladius species based on the combination of the midlegs without tibial comb and with a reduced tibial spur, tapering anal point, and bi-lobed inferior volsella, together with gonostylus with a gentle curve, with a tip directed postero-laterally.

Description

Habitus: Total length 1.2–1.7 mm. Overall light yellowish-brown coloration, with thorax and pedicelli darker than the rest of the body.

Head: Eyes bare, kidney-shaped, without dorsomedial extension. Palpomeres (2–5) length in µm (n = 2, No. OU47572, No. OU47575): 23, 48–60, 80, 92–93 (Fig. 5A). Clypeus square, with at least eight setae (paratype No. OU47575). Palpomere three with a possible small distal protrusion, but condition of specimens not permitting corroboration of that. Antennae with 13 flagellomeres, (flagellomeres measurable on holotype only, length in μm): Fm1: 14, Fm2: 16, Fm3: 25, Fm4: 25, Fm5: 19, Fm6: 21, Fm7: 21, Fm8: 17, Fm9: 15, Fm10: 24, Fm11: 15, Fm12: 23, Fm13: 166, AR = 0.7.

Thorax: Acrostichals setae 5–8, strong and decumbent, starting close to the antepronotum, getting larger towards the posterior. In holotype No. OU47576: 5 visible, paratype No. OU47540: 8 visible. Dorsocentrals 7, uniserial; scutellars 8, uniserial. Postnotum bare.

Legs: Leg segments lengths as listed in Table 1. Terminal tarsomeres without pulvilli, shape of all flagellomeres cylindrical. Foreleg tibial spurs 10–31 μm (n = 2), midlegs without tibial comb and with a reduced tibial spur (Fig. 2D), hindtibia with two spurs, short 6–16 μm (n = 3), long 17–31 μm (n = 3) hindtibia comb made of 7–9 (n = 3) setae. Spurs with very weak lateral denticles, compressed to the main spur’s shaft (Fig. 5A1). Tarsomeres without pseudospurs.

Wings: 0.71–0.96, mean = 0.83 mm long (n = 5). Anal lobe strongly reduced. Costal extension ca. 85 µm long (n = 1). Cu1 slightly sinuate. Squama fully fringed, with at least 11 setae (n = 1, holotype) (Figs. 3A, 3B). Wing membranes without macrotrichia, with coarse punctuation. Venation as in Fig. 3A.

Hypopygium: Anal point long, expanding distally, bare, 23–40 μm long (n = 3), parallel-sided for the most of the length, widening distally. Gonocoxite 60–140 μm long (n = 3), with a large, bi-lobed inferior volsella, consisting of larger, anvil-shaped ventral lobe, and smaller, finger-like lobe, directed medio-posteriorly (Figs. 4A–4D, 7C, 7D). Gonostylus with a gentle curve, with a tip diverging outside (laterally from the body’s midline), 43 μm long (n = 1, holotype). Megasetae present, crista dorsalis absent (Figs. 4A–4D).

Taxonomic notes

The new species belongs to the genus Bryophaenocladius based on the combination of bare eyes, bare wings, fringed squama, lateral spines compressed to the shaft of the tibial spurs (note that spines are very weak, but not dissimilar from B. muscicola Kieffer, 1906 or B. chrissichuckorum Epler, 2012), pulvilli absent, acrostichal setae strong and decumbent, comb present on the hindtibia, and anal point well developed (Cranston, Oliver & Sæther, 1989). In the absence of a modern, comprehensive revision of the genus Bryophaenocladius it is difficult to ascertain relations between the new fossil species and other species of Bryophaenocladius. The general shape of the hypopygium, particularly the long, distally expanding anal point, is highly reminiscent of B. beuki, Baranov, Andersen & Hagenlund, 2015 from Baltic amber. Among extant taxa, the hypopygium of the new species is quite similar to B. psilacrus Sæther, 1982 in the bi-lobed inferior volsella, with larger anvil-shaped ventral lobe and a smaller, finger-like dorsal lobe, the long anal point, and the gently curving gonostylus, without crista dorsalis, as well as to B. vernalis (Goetghebuer, 1921) (Brundin, 1956; Makarchenko & Makarchenko, 2006).

Morphotype 1 cf. Bryophaenocladius zealandiae

(Figs. 1C, 8, Table 2)

Figure 8 Morphotype 1 cf. Bryophaenocladius zealandiae.

(A) Habitus of specimen OU47574. (B) Habitus of specimen OU47573. (C) Palpomere 3 (OU47574), arrow marks distal protrusion. (D) Hypopygium, lateral view (OU47574).

Table 2 Length (in μm) of leg segments of morphotype 1 cf. Bryophaenocladius zealandiae (measured on two specimens).

Leg	Femora	Tibia	Ta1	Ta2	Ta3	Ta4	Ta5	
Foreleg	471–518	461–606	325–415	111–115	101–109	72–136	46–79	
Midleg	409–511	471–561	178–250	93–144	65–117	33–49	50–51	
Hindleg	489–565	530–570	293–340	153–189	95–142	52–79	62–72	
Note:

Values are given as min–max range.

Material. No. OU47573 and No. OU47574, both complete and fairly well visible within yellowish-orange translucent amber.

Description

Habitus: Total length 2 mm, wing length 1 mm (n = 2). Colour: dark brown head and body, and legs of a lighter-brown colour.

Head: Eyes bare, kidney-shaped, without dorsomedial extension. Palpomeres (2–5) length in µm (n = 2): 26, 45–77, 46–52, 83–100 (Figs. 7A–7D). Clypeus square. Palpomere three with a distinct conical protrusion on the distal end.

Thorax: Acrostichals setae strong and decumbent, 8 present (n = 2). Dorsocentrals present but difficult to count, uniserial, at least 4. Postnotum bare. Anepisternum and epimeron without leaf-shaped setae.

Legs: Leg segments lengths as listed in Table 2. Terminal tarsomeres without pulvilli, shape of all the flagellomeres cylindrical. Foreleg tibial spurs 16 μm (n = 2), midlegs without tibial spur, hindtibia with two spurs, short 27 μm (n = 1), long 30–46 μm (n = 2) hindtibia comb made of 6–8 (n = 2) setae. Tarsomeres without pseudospurs. Lateral spines compressed to the shaft of the tibial spurs.

Wings: 1 mm long (n = 2). Details of venation not observable.

Hypopygium: Only visible in lateral aspect. Anal point bare, ca 50 μm long (n = 1). Gonocoxite ca. 150 μm long (n = 1).

Taxonomic notes

This morphotype is similar to Bryophaenocladius zealandiae sp. nov.; however, wing venation and structure of terminalia are not decipherable. Should they indeed be members of Bryophaenocladius zealandiae sp. nov., this will corroborate that this species has a distal projection on the end of palpomer 3, supporting affinity with the subgenus Odontocladius Albu, 1974.

Genus Pterosis Sublette & Wirth, 1980

Amended generic diagnosis: Can be distinguished from any other Chironomidae based on the combination of absent apical setae on the last flagellomere of a male, the well-developed antepronotum, scutellars biserial, wing membrane fully covered with macrotrichia, R2+3 vein reaching almost to the apex of the R1 vein, squama fringed, inferior volsella with various density of setae, anal point heavily setose, gonostylus with a crista dorsalis of variable size.

Pterosis extinctus sp. nov. Baranov

(Figs. 1C, 9, 10, Table 3)

Figure 9 Pterosis extinctus sp. nov. Baranov, holotype OU47546, male.

(A) Habitus. (B) Head. (C) Last flagellomere’s a crown of gentle sensillae marked by arrow. (D) hypopygium , ventral view. Abbreviations: Aps, antepronotal setae; H, humeral setae.

Figure 10 Pterosis extinctus sp. nov. Baranov, holotype OU47546.

(A) Photomicrograph of wing. (B) Line drawing of wing. (C) Photomicrograph of hypopygium, dorsal view. (D) Line drawing of hypopygium, dorsal view. Abbreviations: AP, anal point; Gc, gonocoxite; Gs, gonostylus; IVo, inferior volsella; TIX, abdominal tergite 9; C, costal vein; Cu1, cubital vein 1; M1+2, medial vein 1+2; M3+4, medial vein 3+4; R1, radial vein 1; R2+3, radial vein 2+3; R4+5, radial vein 4+5.

Table 3 Length (in μm) of leg segments of Pterosis extinctus sp. nov. Baranov, male holotype No. OU47546.

Leg	Femora	Tibia	Ta1	Ta2	Ta3	Ta4	Ta5	
Foreleg	704	741	423	175	222	161	217	
Midleg	596	520	468	124	76	–	–	
Hindleg	615	783	–	–	–	–	–	

Zoobank LSID: urn:lsid:zoobank.org:act:DBC38BCD-7C3A-489A-8B47-344576745488

Holotype. No. OU47546; male, partially preserved (thorax is missing), in a piece of semi-translucent, yellow amber (7 × 5 × 0.5 mm) with abundant small air-bubbles (Figs. 1C, 9, 10).

Derivation of name. After Latin “extinctus”, meaning extinct.

Type locality and horizon. Temporary lignite pit, site G45/f0107, near Tapanui, southern New Zealand; Pomahaka Formation, late Oligocene (Chattian, New Zealand stage Duntroonian).

Diagnosis. This fossil species can be distinguished from the only other known Pterosis species, Pterosis wisei Sublette & Wirth, 1980, based on the combination of the following characters: apical flagellomere with a crown of sensillae, antepronotal setae present close to the midlength of the antepronotum, Cu1 slightly sinuate, inferior volsella lightly setose, anal point blunt and setose, gonostylus with weak crista dorsalis.

Description

Adult male (No. OU47546)

Habitus: Total length 2.3 mm. Colour: dark brown across the parts of the body.

Head: Eyes bare, presence of the dorsomedial extension impossible to ascertain. Palpomeres (3–5) length in µm (n = 1): 115, 126, 143 (Fig. 8B). Clypeus square with at least 10 setae. Antennae with 13 flagellomeres, (flagellomeres measurable on holotype only, length in μm): Fm1: 27, Fm2: 30, Fm3: 19, Fm4: 31, Fm5: 30, Fm6: 31, Fm7: 25, Fm8: 30, Fm9: 27, Fm10: 28, Fm11: 28, Fm12: 31, Fm13: 360, AR = 1.1 Flagellomere 13 with a crown of gentle sensillae (Fig. 9C).

Thorax: Most of the thorax, except for antepronotum and part of the scutum, missing. Antepronotal lobes well developed, meeting medially. Strong antepronotal setae present (at least two), reaching the mid-length of the antepronotal lobes. Small piece of mesonotum still preserved (Fig. 10A), strongly projecting forward, over the head. Three humerals visible (Fig. 9B).

Legs: Leg segment lengths as listed in Table 3. Foreleg tibial spurs 30 μm (n = 1), presence and number of other spurs impossible to ascertain. Presence of the pseudospurs on the foreleg tibia impossible to ascertain, due to it being surrounded by a dense cloud of bubbles. Pulvilli absent.

Wings: 1.5 mm long (n = 1). Wing membrane densely covered with macrotrichia. Cu1 slightly sinuate, costal extension produced slightly beyond R4+5 insertion, otherwise, venation as in Fig. 10B. Squama invisible (Figs. 10A, 10B).

Hypopygium: With numerous long setae, gonocoxite 114 μm long (n = 1). Gonostylus ca. 70 μm long (n = 1), expanding distally, without obvious crista, but with sub-oval expansion ventrally, megasetae short and sturdy. Anal point short, cresting top of tergite IX, with several long setae (Figs. 10C, 10D). Inferior volsella subrectangular, narrowing distally (Figs. 10C, 10D). Presence of virga impossible to ascertain, but since hypopygium is partially transparent, we can rule out presence of extremely strong and sclerotized virga.

Taxonomic notes

This species is attributed to the genus Pterosis based on the combination of bare eyes, apical flagellomere without subapical sensillae, wing fully covered with macrotrichia with costal extension, antepronotals present, mesonotum strongly projected forward over the head, humerals present, presence of crest-like anal point on tergite IX, absence of virga and overall extremely high density of setae on the body (Sublette & Wirth, 1980). Since the hypopygium and tergite VIII are partially translucent in the male specimen, and the extremely large and sclerotized virga is not visible, its presence is unlikely. Absence of the apical setae on the 13th flagellomere differentiates this species from representatives of Gymnometriocnemus Edwards, 1932a (Sublette & Wirth, 1980; Sæther, 1983; Stur & Ekrem, 2015). The species can be differentiated from representatives of Allometriocnemus Freeman, 1961, by combination of the lobes of antepronotum meeting medially and wing membrane being completely covered with macrotrichia (Freeman, 1961; Sublette & Wirth, 1980).

P. extinctus can be easily differentiated from P. wisei by antepronotal setae of the former being closer to the midlength of the antepronotum, in contrast to P. wisei, whose antepronotals are all concentrated on the distal part of the antepronotum, as well as much smaller crista dorsalis of the new species (Sublette & Wirth, 1980).

Morphotype 2 cf. Metriocnemini

(Figs. 11, 12, Table 4)

Figure 11 Morphotype 2, cf. Metriocnemini, females.

(A, B) Habitus and wing of specimen OU47577. (C) Habitus of specimen OU47578.

Figure 12 Morphotype 2, cf. Metriocnemini, females.

(A) Head of specimen OU47577. (B) Head of specimen OU47578; B1 close-up of last flagellomere with apical setae. (C) Female genitalia (OU47578). (D) Female genitalia, marked (OU47578). Abbreviations: A8, abdominal segment 8; Ce, cerci; Gca, gonocoxite apodem; Gc, gonocoxite (8); Gp8, gonapophysis 8; TIX, tergite 9.

Table 4 Length (in μm) of leg segments of Morphotype 2, females (measured on different numbers of specimens, depending on the preservation of leg segments).

Leg	Femora	Tibia	Ta1	Ta2	Ta3	Ta4	Ta5	
Foreleg	312	308	109–139	56–70	48–50	28–34	26–46	
Midleg	238–282	259–266	92–100	39–49	33–34	23–29	38–47	
Hindleg	267–269	211–296	145–171	55–69	70–89	37–40	42–50	
Note:

Values are given as min–max range.

Material. No. OU47577 and No. OU47578, adult females, both complete and fairly well visible within yellowish-orange translucent amber.

Description

Adult female

Habitus: Total length 0.9–1.0 mm. Colour: dark brown across the parts of the body.

Head: Eyes bare, reniform. Palpomeres (2–5) length in µm (n = 2): 29 (n = 1, OU47578), 40–41, 41–42, 75–82 (Figs. 11A–11C, 12A, 12B). Clypeus square with at least 11 setae. Antennae with 5 flagellomeres, (n = 2, length in μm): Fl1: 76 (n = 1, OU47577), Fl2: 19–29, Fl3: 26–28, Fl4: 22–27, Fl5: 39–40. Flagellomere 5 with a weak but distinct subapical seta (Figs. 12A, 12B). Pedicellus cup-shaped.

Thorax: Acrostichals setae strong and decumbent, 5–8 (n = 2). Dorsocentrals biserial, upper row 5, lower row 8. Postnotum bare. Antepronotum 4. Anepisternum and epimeron without leaf-shaped setae. Prealars 3, humerals 4. Scuterals uniserial, 6.

Legs: Leg segments lengths as listed in Table 4. Foreleg tibial spurs 14–15 μm (n = 2), midtibial spur 11–15, hindtibial spur 26–37 (length in μm). Hindtibial comb made of 8–9 strong seate. Pulvilli absent, empodium feathery.

Wing: 0.63–0.72 mm long (n = 2). Wing membrane densely covered with macrotrichia. Cu1 slightly sinuate. Squama bare, costal extension pronounced (Fig. 11C). Wing with numerous macrotrichia, otherwise as shown on the Fig. 11B. Halters dark-brown in their entirety.

Female genitalia: Cerci very small, gonapophysis VIII divided into small mesal lobe and narrow dorsomesal lobe (Figs. 12C, 12D). Gonocoxite relatively small, with at least 5 strong setae. Tergite IX rounded, undivided (Figs. 12C, 12D).

Taxonomic notes

Dense macrotrichia of the wings, as well as dense setation of the thorax, with antepronotals present, is indicative of a close affinity of this morphotype with representatives of the genera Metriocnemus van der Wulp, 1874 or Gymnometriocnemus, with more precise determination of taxonomic affinity impossible without additional material (Sæther, 1977).

Discussion

Faunal affinities and biogeography

Chironomids have a long history, with the oldest representatives occurring in the uppermost Triassic of Europe (203 mya) (Krzemiński & Jarzembowski, 1999), although based on dated phylogenies the group is likely significantly older, at least 250 mya (Cranston, Hardy & Morse, 2012). The oldest Orthocladiinae fossils of Lebanorthocladius furcatus Veltz, Azar & Nel, 2007 are known from Lower Cretaceous Lebanese amber (Veltz, Azar & Nel, 2007). The long geological history and rich fossil record has made chironomids a suitable model group for historical biogeographic analyses. Following Hennig’s (1966) work on phylogenetic systematics, Lars Brundin became interested in applying principles of cladistic analysis and an emerging understanding of plate tectonics to the analysis of Chironomidae distribution in the Southern Hemisphere (Brundin, 1966). Brundin came to the conclusion that the majority of Chironomidae distribution patterns in Australia, Southern Neotropics and New Zealand can be explained by the break-up of Gondwana. Since then, however, our understanding of the assembly of New Zealand’s biota has become more refined. In particular the role of dispersal has become more widely accepted (e.g., Trewick, 2000; Sanmartin, Enghoff & Ronquist, 2001). The composition of the New Zealand Chironomidae, particularly the Orthocladiinae fauna, reflects a complex history influenced by both trans-Tasman and trans-Antarctic dispersal and vicariance following the break-up of Gondwana (Krosch & Cranston, 2013; Krosch et al., 2011, 2015).

Bryophaenocladius has a near worldwide distribution, although there are only preliminary records from Australia (Cranston, 1996) and the genus is seemingly absent from the extant fauna of New Zealand (Boothroyd & Forsyth, 2011; Ashe & O’Connor, 2012). This study follows on the specimens from Oligocene amber, preliminarily classified by us into genus Bryophaenocladius in our earlier article (Schmidt et al., 2018), which was the first record of Bryophaenocladius from New Zealand. We also noted that the BOLD V4 system has barcoding records of Bryophaenocladius in New Zealand (Schmidt et al., 2018) but, on closer examination, these belong to the two BOLD BINs BOLD:AAM6273 and BOLD:AAG1021. Representatives of these BINs all cluster around the Holarctic species Bryophaenocladius ictericus (Meigen, 1830). It is thus likely that this species has been historically introduced to New Zealand (and Australia) with agricultural produce, as Bryophaenocladius larvae are associated with agricultural plants (Cranston, 1987).

While there appear to be no native species of Bryophaenocladius on the main islands of New Zealand, it is highly likely that the monotypic Kuschelius dentifer Sublette & Wirth, 1980, endemic to the sub-Antarctic Auckland Islands, is in fact a species of Bryophaenocladius. Sublette & Wirth (1980) erected the genus Kuschelius as intermediate between Chaetocladius Kieffer, 1911 and Bryophaenocladius, and distinguished K. dentifer from species of Bryophaenocladius by the presence of apical setae on the terminal flagellomere of the antenna and slightly diverted spines on the tibial spur of the hind leg (Figs. 13A–13E). However, these characters in combination with the structure of the hypopygium and the presence of the distal projection on the distal end of palpomere 3 fit well within the current definition of Bryophaenocladius, subgenus Odontocladius Albu, 1974 (Albu, 1974; Armitage, 1987; Moubayed & Langton, 2023). As pointed out by Sæther (1982) and Armitage (1987), K. dentifer is almost certainly a Bryophaenocladius, very similar to B. brincki (Freeman, 1955) originally described from South Africa. Molecular data on K. dentifer are not yet available. B. zealandiae sp. nov. Baranov from Pomahaka amber now confirms the occurrence of the genus Bryophaenocladius in Zealandia in the late Oligocene (~26 mya) and documents its post Oligocene extinction in New Zealand, at least on the main islands. The only two previously reported fossils of Bryophaenocladius (B. beuki Baranov, Andersen & Hagenlund, 2015 and B. circumclusus Seredszus & Wichard, 2007) are from Eocene Baltic amber. A review of modern Bryophaenocladius and additional fossils along with further fossil specimens would be of a great importance to deciphering the biogeographic history of this genus.

Figure 13 Holotype (adult male, number NZAC02044947) of Kuschelius dentifer Sublette & Wirth, 1980.

(A) Wing. (B) Head, arrow marks an apical protrusion of the 3rd palpomere. (C) hypopygium. (D) Midtibia with spurs and the comb. (E) Hindtibia with the spurs and comb. All photos in this plate are made by Dr. Leanne Elder, licensed under CC BY 4.0 and used with the photographer’s explicit permission.

The genus Pterosis identified here from Pomahaka amber includes one extant species, P. wisei Sublette & Wirth, 1980, endemic to the subantarctic Auckland Islands and Campbell Island of New Zealand (Sublette & Wirth, 1980). The discovery of Pterosis extinctus sp. nov. Baranov in amber from the Pomahaka Formation shows the presence of Pterosis on mainland New Zealand in the late Oligocene.

Paleoecology of non-biting midges from Pomahaka amber

It is notable that both newly discovered species of midges from Pomahaka amber belong to Chironomidae groups whose extant representatives have larvae that develop mostly in terrestrial and semi-aquatic habitats (Moller Pillot, 2013), and not in the aquatic habitat seen in larvae of most other chironomids. Larvae of Bryophaenocladius develop in wet mosses, decaying leaves or similar wet habitats (Strenzke, 1957; Moller Pillot, 2013). Larvae of Pterosis are unknown, but given the adults’ similarity to Gymnometriocnemus representatives, larvae of Pterosis likely develop in wet terrestrial habitats as well (Sublette & Wirth, 1980). Terrestrial and semi-terrestrial Chironomidae are relatively common in various amber deposits worldwide, probably due to their association with mosses and other microhabitats on the bark of the resin-producing trees or on the nearby forest floor (Solórzano-Kraemer et al., 2018). Prior to the research presented in this article, two fossil species of Bryophaenocladius were known: B. beuki Baranov, Andersen & Hagenlund, 2015 and B. circumclusus Seredszus & Wichard, 2007 and a probable larva of this genus (Baranov et al., 2019), all from Eocene Baltic amber. Until now no fossil Pterosis were known but there are numerous other fossils of Chironomidae whose extant representatives develop in terrestrial habitats, such as Parametriocnemus Goetghebuer, 1932, Paraphaenocladius Spärck & Thienemann, 1924, Pseudorthocladius Goetghebuer, 1943, Smittia Holmgren, 1869 and Pseudosmittia Edwards, 1932b (Zelentsov et al., 2012; Baranov, Andersen & Hagenlund, 2015). The prevalence of the Chironomidae with terrestrial larvae in certain amber deposits indicates high humidity in amber forest habitats, as relatively high and constant humidity is required by these groups of Chironomidae to finish larval development (Strenzke, 1957; Armitage, Cranston & Pinder, 1995, Zelentsov et al., 2012). The finding of terrestrial or semi-aquatic midges in Pomahaka amber is consistent with the paleo-environmental reconstruction. The amber-bearing lignites of Pomahaka Formation formed by in-situ growth and decomposition of wetland forest trees and litter in domed forest swamps (Lindqvist, Gard & Lee, 2016) and the palynomorph assemblage from the lignites includes ferns, shrubs, herbs and reeds associated with moist and damp habitats which indicates high humidity and high rainfall throughout the year (Pocknall, 1982).

We sincerely thank Andrew Morris for allowing access to his property and for generous help with excavating Pomahaka amber. The Geology Department, University of Otago is acknowledged for help with fieldwork logistics and for curating fossil specimens in the collections of the Geology Museum. We are grateful to Leanne Elder, New Zealand Arthropod Collection, Manaaki Whenua—Landcare Research, for taking photos of Kuchelius dentifer, and allowing us to use those photos in this publication. We thank Dagmara Żyła—handling editor of the article and the reviewers, including Armin Namayandeh and two anonymous reviewers, for constructive and detailed comments.

Additional Information and Declarations

Competing Interests

The authors declare that they have no competing interests.

Author Contributions

Viktor O. Baranov conceived and designed the experiments, performed the experiments, analyzed the data, prepared figures and/or tables, authored or reviewed drafts of the article, and approved the final draft.

Jörg U. Hammel performed the experiments, prepared figures and/or tables, authored or reviewed drafts of the article, and approved the final draft.

Daphne E. Lee performed the experiments, authored or reviewed drafts of the article, and approved the final draft.

Alexander R. Schmidt conceived and designed the experiments, performed the experiments, analyzed the data, authored or reviewed drafts of the article, and approved the final draft.

Uwe Kaulfuss conceived and designed the experiments, performed the experiments, analyzed the data, prepared figures and/or tables, authored or reviewed drafts of the article, and approved the final draft.

Data Availability

The following information was supplied regarding data availability:

All specimens included in this work are housed at the Geology Museum of the Geology Department, University of Otago (OU): Bryophaenocladius zealandiae sp. nov. Baranov, OU47576 (holotype); OU47540, OU47572, OU47573, OU47574, OU47575 (paratypes); OU47579, OU47580, OU47581, OU47582 (associated material). Morphotype 1, cf. Bryophaenocladius, OU47573, OU47574. Gymnometriocnemus (G.) extinctus sp. nov. Baranov, OU47546 (holotype); OU47577, OU47578 (paratypes).

The dataset including the original tiff-images (scaled-down as described above), pvl.nc file and a movie of the specimens are available at zenodo repository on 10.5281/zenodo.14517316.

New Species Registration

The following information was supplied regarding the registration of a newly described species:

Publication LSID: urn:lsid:zoobank.org:pub:1B39CC4B-AA24-4D9F-819B-F1834E615C5C.

Bryophaenocladius zealandiae LSID: urn:lsid:zoobank.org:act:2FE3D4EA-5F0B-4BB4-A2CC-E88557CE825D.

Pterosis extinctus LSID: urn:lsid:zoobank.org:act:DBC38BCD-7C3A-489A-8B47-344576745488.

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
