# Peer review of "Extending the fossil record of late Oligocene non-biting midges (Chironomidae, Diptera) of New Zealand"

_PeerJ, doi:10.7717/peerj.18893_

## Round 0.1 · original submission · Major Revisions

Please, address all the reviewers' suggestions and comments. There are some concerns raised over the taxonomic justification.

Reviewer 1 ·

Basic reporting

Review of the manuscript "Extending the fossil record of late Oligocene non-biting midges (Chironomidae, Diptera) of New Zealand” for PeerJ.

The article presents very important findings, is interesting and well done. The new data contained in the manuscript are very valuable for knowledge. The research methods are correct. All figures are all suitable for this manuscript and are well done. The manuscript should be published as soon as possible.
However, I suggest a minor revision.
- line 38: I suggest adding the age of Pomahaka amber;
- e.g. lines 153, 167: It would be better to add acronym of the place where the material was deposited. This is given in Matherial and methotds, but will be better for the reader. It woula also be clearer to add the abbreviation „No.” before all numbers
[correct this consistently throughout the work];
- line 220, 223: „hypopigium” was used, it should be „hypopygium”, this term was used 21 times in this manuscript (lines 305, 340, 393, 719, 730, 731, 738, 741, 745, 746, 764, captions of figures: 5, 7, 8, 9, 10, 13
[correct this consistently throughout the work];
- all new taxa must be registered in Zoobank;
- please check the English, spelling and formating, e.g. lines 501, 510, 516 and others, you have a dot at the end of the citation, in some cases lines 522, 524, 626 are without;
- Figure 12D – the quality needs to be improved, I mean the edges of the bristles where the background is not precisely cut out.

Sincerely yours,

Experimental design

no comment

Validity of the findings

The new data contained in the manuscript are very valuable for knowledge.

·

Basic reporting

I am delighted to read this article and have learned a lot in the process. The authors have worked hard to write this article, and their efforts and results are commendable. It is for this reason that I would like them to consider improving their article further, as I very much like to see this study published. The specific comments are provided in the attached documents. Here, I will stress the following that were repeatedly noticed in the article and need revision.
1. At first mention, full species name with authority and date. After that, only abbreviated.
2. Please pay attention to the spelling of the species and ensure proper authorities are cited.
3. Keep your sentencing short; divide long sentences into smaller ones as much as possible.
4. Using simple language and words instead of complex ones.
5. Please pay attention to your sentence and how it is structured. See my comments in the attached file.
6. Please provide DOI for all references, if available. Also, double-check the format
7. In taxonomic description, it is better and easier for readership if you do the following
• Palpomeres (x-y) length in µm: xx, xx, xx, xx
• Achrostichals x, strong and decumbent; scutellars 8, uniserial
• Please avoid duplicating the measurements.
• Remove citations from the diagnosis section. Please also remove the discussion of comparison to other similar species from the diagnosis section. These should be in the remarks section.

Experimental design

No comment

Validity of the findings

No Comment

Reviewer 3 ·

Basic reporting

The manuscript includes sufficient introduction and background based on the previous work and relevant literature concerning the fossil record, as well as extant distribution of Orthocladiinae midges.

The structure of the article is mostly consistent with the Journal' Instructions for Authors: text formatting, figures and tables are prepared adequately, with the need for minor refinement (see comments below).

English needs to be improved throughout the manuscript, as in some parts punctuation, word usage and overall sentence construction may present a challenge for the future Reader.

Figures are relevant to the content of the article and of sufficient resolution, some descriptions and labels needs improvement (see comments below).

The manuscript is ‘self-contained' and represents an appropriate ‘unit of publication’, but it lacks some of the data relevant to support the decisions made in it. The concepts of erection of new species need to be reassessed by taking into account the Reviewer' comments and suggestions (see comments below).

Experimental design

Presented research is in line with aims and scope of the Journal.

Research question were well defined and relevant. Authors specified how the research fills the knowledge gap.

The research have been conducted to a high technical standard and in conformity with the prevailing ethical standards in the field. All methods were described with enough detail to replicate.

Validity of the findings

The study presents the two new fossil species of the family Chironomidae, that have been identified in Pomahaka amber from the late Oligocene of New Zealand. They represent the subfamily Orthocladiinae, which is characterised by a high level of diversity and endemism in the region.
A new species, Bryophaenocladius zealandiae sp. nov., was identified, representing the first fossil record of the genus in the Southern Hemisphere. Two additional specimens bear resemblance to the new species, suggesting possible intraspecific size variation or the presence of another similar taxon. The second identified species, Gymnometriocnemus extinctus sp. nov. represents the genus which is currently known to possess a single endemic species in New Zealand. These newly discovered fossil taxa are also indicative of a terrestrial or semi-aquatic habitat, suggesting a humid palaeoenvironment for the Pomahaka amber forest. The discovery offers valuable insights into the evolutionary history and potential migration patterns of this group within New Zealand, as well as constituting a significant contribution to a more comprehensive understanding of the biogeographical history of the continent. Simultaneously, it serves as an illustration of the unique and valuable insights that can be derived from the study of inclusions in amber.

Still, as I stated above in Section 1 and in comments below, the manuscript lacks some data relevant to fully support its decisions. The concepts of erection of new species need to be reassessed.

Additional comments

hypopygium – misspelled numerous times as “hypopigium” (lines: 220, 223, 305, 340, 393, 719, 730, 731, 738, 741, 745, 746, 764)

'References' section: not every reference is placed correctly in an alphabetical order, many positions lack DOI

line 35: typo in Gym[n]ometriocnemus

line 56: should be „on the South Island”

line 67: The term "non-biting midges" is equivalent with the whole family, whereas your construction of the sentence might suggest that it covers subfamily Orthocladiinae alone.

lines: 76-77: link provided in References section (line 556) does not lead to the given location.

line 95 and throughout the manuscript: please provide the full specific names when the taxon is mentioned the first time in your manuscript, along with the authority and the year of publication.

line 140: The work of “Schindelin et al., 2012” is not included in the References section

lines 144-150: those references are not in chronological order

line 147: typo – it should be “Kaczorowska & Giłka (2002)”

line 157: typo – it should be “Newman, 1834”

lines 166-168: Figures should be arranged in the order in which they appear it the text. Therefore, please move the paratypes’ figure behind those 3 that covers the holotype.

lines 175-180: please provide a more strongly supported differential diagnosis, by highlighting the differences between your species and the type species of the genus. Additionally, in the diagnosis section is a long series of references (they are also not in a chronological order) - it is not entirely clear what they are referring to… As only differentiating/distinctive features should be included in the diagnosis, please move them and discuss them perhaps in the section “Taxonomic notes”?

line 185: Correct abbreviation of palpomere (after Saether 1980) is Pm1-5 . Also palpomeres are not visible on the given figure (Fig. 2A).

lines 188-189: Correct abbreviation of palpomere (after Saether 1980) is Fm1-13. Also, is there a reason why first flagellomere measurement is given to 3 decimal places while others not?

lines 191-194 and 195-199: in both ‘Thorax’ and ‘Legs’ sections the references to relevant figures would be appreciated

line 191: Please, specify whether acrostichals start closely to anterior margin of scutum, as it is one of important characters in determining the correct generic affiliation of the species

lines 196-197: Are midlegs with spurs absent or is it possible, that they are not visible? If they are truly absent in all examined specimens – it would be an interesting character, worth putting in the species’ diagnosis

line 198-199: “Lateral spines compressed to the shaft of the tibial spurs” – it is an important character, manuscript would benefit from the addition of a photograph and/or drawing

line 203: please change ‘elements’ to ‘segments’, as leg elements may refer to every part of it, not only segments

Table 1: all numbers in leg segments’ abbreviations should be written in lower index

line 207: incorrect Figure references, it should be ‘Fig. 4A, B’. Also, please complete this section with a note: “Venation as in Fig. 4 A

lines 209-210: concerning inferior volsella: What about the finger-like, medio-posteriorly directed appendage, that is visible from the both sides on both photos (Fig. 5A, C) and drawing (Fig. 5B)? Its position and shape excludes it from being a part of axe-shaped inferior volsella…

line 210: “gonostylus gently curved distally” please precise in what direction

line 217: “anal point hyaline” – I would discuss it being hyaline – for an amber specimen it looks as quite strongly sclerotized, especially when compared with semi-translucent gonostyli (I am reffering to the Fig. 5A)

line 235: Are total length and wing length are based on one or two examined specimens? Please specify.

line 250: there is no mean value in the Table 2.

lines 253-255: How those specimens are 30-50% larger than a new species? The difference between the largest measurements of the new species and those of morphotype(s) is 17.5% in total length (1.7 vs 2.0 mm), and only 4% when comparing wing length ( 0.96 vs 1.0 mm), and are distinctly different from those given by Authors. I noticed that some leg segments are significantly longer, but then other are shorter comparing with new species’ type series. Taking into consideration the whole legs’ lengths – they are fitting quite nicely with the specimens you consider as the type series for B. zealandiae sp. nov. Still, I understand and support your concept of treating those two specimens as morphotypes, especially with wing venation not visible and hypopygium possible to check only from lateral, but I suggest you rephrase the “Taxonomic notes” section.

line 268: ”Latin” should be written with upper letter; a noun “extinct” in Latin is “ex[s]tinctus”, you have confused it with a verb “ex[s]tinguo”.

line 290: please, specify if it is a single seta or plural setae, as for the genus Gymnometriocnemus the diagnostic feature is the presence of only one strongly elongated seta. On Fig. 9C visible subapical setae are more-or-less comparable in length, what makes me concerned about the correctness of the attribution.

line 294: Please add information if pseudospurs on tarsi are present/absent/unobservable – it is quite an important character.

line 300: on the photo and drawing costal extention does not look pronounced – please, verify it. Next issue: you wrote “squama apparently bare” and gave references to check it on Fig. 10A, B. I cannot see squama on those figures.

lines 308 and 348: Authors decided to associate two females with the newly designated species. I can see a few issues:
• Firstly, those females are less than half the size of males of newly designated species. In Chironomidae its usually females who are slightly bigger, but here the difference in size is so significant that it is difficult to even consider the compatibility of copulatory organs.
• Secondly: Authors based this decision on a similarity of wing venation and antepronotum setation. Unforunately, the Fig. 11 is not informative enough, as only cubital region of the wing is visible. Therefore, the addition of a female wing drawing will be crucial.
• Thirdly: Authors use term ‘tentatively’ twice, what means that they are also not convinced that those females represent the species. I would suggest to present them without affiliation, but as a specimens similar (cf.) to the species.

line 344: in this situation it is not a synapomorphy, but an autapomorphy

line 356: there is an error - in the cited article Ansorge described species dated to Toarcian (Early Jurrasic), while the discovery of the oldest Chironomidae (Rhaetian, Late Triassic) belongs to W. Krzemiński and E. Krzemińska (1999)(Aenne triassica sp. n., the oldest representative of the family Chironomidae (Insecta: Diptera). Polish Journal of Entomology 68:445-449.)

line 357: missing refference or erroneus date: Cranston et al., 2011

lines 359, 431, 433: typo “Chironomids” - in the middle of the sentence it should either be “chironomids” or “Chironomidae”

line 361: missing refference or erroneus date: Hennig, 1959

lines 372-374: You wrote “The two new species from Pomahaka amber are among the first fossil records of Orthocladiinae from New Zaeland” – can you list and provide references to the others findings?

line 505: the citation is incomplete, it lacks fragment: “and National Museum of Ireland”

line 515: R.C. Brazier is an Author of drawings in this publication, but is not considered a 3rd Author

line 519: invalid hyperlink

line 525: it should be “11(1)” instead of “4(1)”

line 533: please remove “Scandinavian Entomology” as it is an unnecessary repetition

line 580: lacks fragment: “Scientific Publication, 64:”

line 609: it should be “314” instead of “312”.

line 624: it should be “20(3)” instead of “20”.

line 635: it should be “13(4)” instead of “13”.

line 654: it should be “Abteilung A: Palaozoologie” instead of “Abteilung A”.

line 674: it should be “24(22)” instead of “24”.

line 676: it should be “48(1)” instead of “48”.

line 685: lacks fragment: “… subdivision” Gore Sheet District (5170).

lines 549-554: Those 2 positions by P. Freeman are only microcited in the manuscript

lines 706, 715-716, 747-749, and on Figures 2, 4, 10: all numbers in used abbreviations should be given in lover index (after Saether 1980)

Figure 7: lack of measurement units over scales

---

## Round 0.2 · Minor Revisions

Please, note that there are still some minor unresolved issues and they should be solved.

Reviewer 1 ·

Basic reporting

The article is well written. The content of the work included very important results. I have no objections. The work can be published as soon as possible.

Experimental design

no comment

Validity of the findings

no comment

Additional comments

no comment

·

Basic reporting

All corrections requested were provided. I only found a single typo in the revised version. Otherwise, the manuscript should be accepted.

Experimental design

No comment

Validity of the findings

No comment

Reviewer 3 ·

Basic reporting

The manuscript includes sufficient introduction and background based on the previous work and relevant literature concerning the fossil record, as well as extant distribution of Orthocladiinae midges.

The structure of the article is mostly consistent with the Journal' Instructions for Authors: text formatting, figures and tables are prepared adequately, with the need for minor refinement (see comments below).

English needs to be improved throughout the manuscript, as in some parts punctuation, word usage and overall sentence construction may present a challenge for the future Reader.

Figures are relevant to the content of the article and of sufficient resolution, some descriptions and labels needs improvement (see comments below).

The manuscript is ‘self-contained' and represents an appropriate ‘unit of publication’, but it lacks some of the data relevant to support the decisions made in it.

Experimental design

Presented research is in line with aims and scope of the Journal.

Research question were well defined and relevant. Authors specified how the research fills the knowledge gap.

The research have been conducted to a high technical standard and in conformity with the prevailing ethical standards in the field. All methods were described with enough detail to replicate.

Validity of the findings

The study presents the two new fossil species of the family Chironomidae, that have been identified in Pomahaka amber from the late Oligocene of New Zealand. They represent the subfamily Orthocladiinae, which is characterised by a high level of diversity and endemism in the region.
A new species, Bryophaenocladius zealandiae sp. nov., was identified, representing the first fossil record of the genus in the Southern Hemisphere. The second identified species, Pterosis extinctus sp. nov. represents the monotypic genus, known to possess a single New Zealand species. These newly discovered fossil taxa are also indicative of a terrestrial or semi-aquatic habitat, suggesting a humid palaeoenvironment for the Pomahaka amber forest. The discovery offers valuable insights into the evolutionary history and potential migration patterns of this group within New Zealand, as well as constituting a significant contribution to a more comprehensive understanding of the biogeographical history of the continent. Simultaneously, it serves as an illustration of the unique and valuable insights that can be derived from the study of inclusions in amber.

Still, as I stated above in Section 1 and in comments below, the manuscript lacks some data relevant to fully support its decisions.

Additional comments

I must request, once again, that you implement the previously recommended amendments, that appear to have been overlooked:

1. Please check spelling and formatting, double spaces, unnecessary commas, etc. – there is still much to improve.
2. Hypopygium – misspelled numerous times as “hypopigium” and is still not corrected in the Figures captions.
3. please provide the full specific names when the taxon is mentioned the first time in your manuscript, along with the authority and the year of publication - the Authors stated that they have corrected it throughout the manuscript, but unfortunately that was done only partially…
4. Figures should be arranged in the order in which they appear it the text. Therefore, please move the paratypes’ figure (currently labelled as Fig. 3) behind those three that truly cover the holotype (currently labelled ad Fig. 4 and Fig. 5), then amend the text accordingly.
5. In the rebuttal letter Authors stated, that they prefer to include microcitation references. Please, be consequent then and include all Authors of taxa mentioned in your manuscript.

Further remarks you can find as a comments placed directly in the attached PDF file.

Annotated reviews are not available for download in order to protect the identity of reviewers who chose to remain anonymous.

---

## Round 0.3 · Minor Revisions

Thank you for addressing all reviewers' comments. The current version is almost ready for publication. The only missing thing is the lack of information about the storage and availability of micro-CT data. The raw data should be available if possible, as well as the reconstructed model should be publicly available at the latest upon publication in generally readable (not software-specific) file formats (Davies et al. 2017) and mentioned within the manuscript. Various repositories are available offering a digital object identifier (e.g., Zenodo or Morphosource or Morphobank), or this could also be the If the institution open repository. Please, make sure the data are accessible.

---

## Round 0.4 · accepted · Accept

Thank you very much for addressing the last comment and making the data accessible. I am happy with the current version and I think it is ready for publication.